# Environmental Stressors and Neuroinflammation: Linking Climate Change to Alzheimer’s Disease

**DOI:** 10.3390/cimb47110959

**Published:** 2025-11-18

**Authors:** Mario Caldarelli, Pierluigi Rio, Antonio Gasbarrini, Giovanni Gambassi, Rossella Cianci

**Affiliations:** 1Department of Translational Medicine and Surgery, Catholic University of Sacred Heart, 00168 Rome, Italy; mario.caldarelli01@icatt.it (M.C.); pierluigi.rio01@icatt.it (P.R.); antonio.gasbarrini@unicatt.it (A.G.); giovanni.gambassi@unicatt.it (G.G.); 2Fondazione Policlinico Universitario A. Gemelli, Istituto di Ricerca e Cura a Carattere Scientifico (IRCCS), 00168 Rome, Italy

**Keywords:** neuroinflammation, environment, climate change, pollution, Alzheimer’s disease

## Abstract

Environmental exposures are widely recognized as major risk factors for human health. According to projections by the World Health Organization, climate change is expected to cause a significant increase in mortality within the next few decades. Environmental factors, including diet, weather, occupational exposures, and pollutants play a key role in human diseases affecting different systems, such as cardiovascular, pulmonary, gastrointestinal, and neurological. This narrative review explores the relationship between environmental stressors and neuropathological mechanisms, such as microglial and astrocytic activation, oxidative stress, and neuronal injury, involved in neuroinflammation and the associated neurodegeneration. The pathogenesis and progression of Alzheimer’s disease is discussed in detail, establishing a link between environmental stressors and neuroinflammation. A deeper understanding of these neuropathological mechanisms may guide the development of preventive and therapeutic strategies to safeguard brain health in the context of global environmental change.

## 1. Introduction

In recent years, increasing attention has been paid to the environment and its potential impact on human health [1]. According to data provided by the Global Burden of Disease (GBD) 2021 study, environmental exposures account for 11 out of 88 risk factors for global health [2]. In 2021, approximately 12.8 million deaths worldwide were attributable to environmental and occupational risk factors [3].

Climate change is a multifaceted phenomenon, resulting from both natural processes and human activities, which significantly contributes to the global burden of diseases and mortality [4]. The 2024 Lancet Countdown report observed a 167% increase in heat-related mortality among individuals aged over 65 years compared with the 1990s, far exceeding what would have been expected in the absence of rising temperatures [5]. In addition, according to projections by the World Health Organization (WHO), between 2030 and 2050, climate change will be responsible for more than 250,000 additional deaths per year, related to heat stress, malnutrition, and diarrhea [6]. Climate change affects different body systems, causing pulmonary, cardiovascular, gastrointestinal, and neurological diseases [1,7].

Both genetic and environmental risk factors contribute to the development of neurodegenerative disorders, such as Parkinson’s disease (PD) and Alzheimer’s disease (AD). It has been established that nutrition, extreme weather phenomena, occupational exposures, and environmental pollutants play a key role in neuroinflammation and neurodegeneration [8]. Environmental and genetic factors converge on common pathogenic mechanisms, with neuroinflammation serving as the critical link between external stressors and neurodegeneration. To clarify how these elements interact, the following section will focus on cellular and molecular processes underlying neuroinflammatory responses.

Neuroinflammation, once thought to be a beneficial response against stressors acting within the central nervous system (CNS), such as infections and injuries, is now considered double-faced [9]. At the cellular and molecular levels, neuroinflammatory responses protect the brain from pathogenic stimuli, but when inappropriately prolonged may result in neurodegeneration [10]. Neuroinflammation arises as the result of elevated levels of cytotoxic molecules and pro-inflammatory mediators within the CNS (e.g., cytokines, prostaglandins and reactive oxygen species, ROS), the stimulation and proliferation of brain-resident cells (e.g., glial cells), the penetration of blood–brain barrier (BBB) by peripheral immune cells (e.g., monocytes and lymphocytes), and, lastly, neurotoxicity with neuronal cell death [11].

In this narrative review, we focus primarily on the detrimental and enduring aspects of neuroinflammation triggered of environmental stressors. While transient neuroinflammatory responses can exert protective effects by promoting pathogen clearance and tissue repair, their chronic activation becomes maladaptive, leading to neuronal damage and contributing to neurodegenerative diseases.

We intend to provide an integrated overview of the cellular and molecular pathways involved in neuroinflammation and how these are affected by environmental stressors. including open and polluted cities, extreme temperatures, and nutrition. Lastly, we will consider the endogenous and exogenous factors contributing to the onset and development of Alzheimer’s disease. We conducted a search on PubMed, MEDLINE, and Google Scholar using the search engine for original and review articles in English published within the last 5 years in peer-reviewed journals.

Article selection was based on predefined inclusion criteria, including study design (original research or systematic reviews), adequate sample size, direct relevance to neuroinflammation and environmental stressors, and methodological quality, as well as broader peer-review status and clarity of statistical analysis. Duplicate studies, along with those that lacked data or employed equivocal methods, were excluded to minimize selection bias.

This review integrates recent data on neuroinflammation at the molecular level with emerging evidence of the impact of environmental stressors and climate change on brain and mental health. This approach aims to further promote research on environment-driven neuroinflammation for future preventive and treatment strategies.

## 2. Neuroinflammation

During past years, inflammation has been considered a byproduct of protein aggregation in the central nervous system. In contrast, novel evidence shows that immune signaling is not just a downstream effect, but an active force driving aggregate formation since the very onset of disease [12].

In homeostatic conditions, immune responses are protective and beneficial to the organism. Once any tissue damage is repaired or an infection is cleared, these responses subside. Instead, when excessively prolonged, inflammation interferes with pro-resolution mechanisms [13,14], leading to chronic inflammation and neurotoxicity.

### 2.1. Core Immune Cells (Microglia, Astrocytes, Crosstalk)

Microglia are the intrinsic macrophages in the central nervous system, serving as its resident immune cells. They serve as the defense mechanism of the CNS, against injury, cancer, age-related neurodegenerative diseases, and stroke [15]. Microglia is responsible for synaptic reduction, regulation of inflammation, phagocytosis, and tissue regeneration, both in healthy and diseased states [16]. Once abnormally stimulated, microglia undergo activation and begin releasing pro-inflammatory cytokines and reactive oxygen and nitrogen species (ROS, NOS), engulfing dying cells, infectious agents, toxic protein aggregates, and cellular debris, while on the other hand releasing also anti-inflammatory cytokines and trophic factors to support resolution and neuronal repair. Nayak et al. underscored the dual protective and inflammatory role of microglia within the CNS [17].

Upon exposure to stressors, microglia exhibit a range of potential activation states that span from pro-inflammatory to anti-inflammatory. Unlike the antiquated M1/M2 paradigm, which assumes a rigid type of association, current findings propose a dynamic, context-dependent spectrum of activation, whereby microglia may express both neurotoxic and neuroprotective features depending on the local cues present [18,19].

Inflammatory microglia are involved in producing cytokines (e.g., TNF-α, IL-1β, IL-6) and generating ROS and NO, and may even prune synapses, thereby contributing to chronic activation that would harm neurons. On the protective side, IL-10 and neurotrophic factors produced by anti-inflammatory microglia, facilitate tissue repair and debris clearance. The equilibrium between these functional states determines whether neuroinflammation is resolved or progresses toward neurodegeneration [20].

Microglia and astrocytes, although differing in lineage, stand anatomically close to each other in the CNS and perform complementary functions [21]. Microglia work as the primary immune responders, surveying the environment, clearing debris, and regulating inflammation; astrocytes, instead, maintain vascular integrity and modulate synapse formation [22]. Both cell types cooperate at pruning synapses and clearing neuronal debris, with microglia engulfing somas and apical dendrites, and astrocytes clearing dendritic apoptotic bodies [13]. Their coordinated actions are orchestrated by molecules, such as Complement component 1q/Complement component 3 (C1q/C3), Multiple EGF-like domains 10 (MEGF10), CD47, IL-33, and are regulated by the receptor tyrosine kinases AXL receptor tyrosine kinase (Axl) and MER proto-oncogene, tyrosine kinase (Mertk) [23,24,25]. Axl and Mertk, expressed in both microglia and astrocytes, mediate phagocytosis during development, as well as in pathological processes, thus creating specific phagocytic territories [23]. In this context, CD47 acts as a “don’t eat me” signal, protecting the synapses from an excessive microglial pruning during brain development [25]. Upon brain damage, astrocytes can promote their microglial phagocytosis, but can further contribute to neuroinflammation by releasing pro-inflammatory cytokines [26]. Overall, a constantly dynamic crosstalk between microglia and astrocytes is imperative for CNS homeostasis, whereas any imbalance can lead to neurodegeneration [27].

Together, these glial interactions coordinate neuroinflammatory processes and regulate intracellular signaling cascades that determine whether the central nervous system shifts toward neuroprotection or neurodegeneration.

### 2.2. Key Signaling Pathways

In physiological conditions, neuroinflammation is initiated, amplified, and resolved through the coordination of several intimately interconnected signaling pathways. Very broadly, three groups can be categorized: (1) transcriptional regulators such as Nuclear Factor kappa-light-chain-enhancer of activated B cells (NF-κB) and Janus Kinase/Signal Transducer and Activator of Transcription (JAK/STAT); (2) stress-activated kinases such as the MAPKs, and (3) inflammasome signaling events by NLRP3.

NF-κB, JAK/STAT, MAPK, and NLRP3 regulatory axes are involved in neurodegeneration.

The NF-κB family transcription factors constitute the fundamental regulators of numerous cellular signaling pathways in the brain [28]. In resting condition, NF-κB dimers remain inactive by associating with inhibitory members of the IκB family, which include IκBα, IκBβ, and IκBε, as well as the precursor proteins p100 and p105 [29]. Activation of NF-kB has been demonstrated in endothelial cells, astrocytes, and microglia, where it mediates the transcription of inflammatory genes and contributes to glial reactivity against stressors [30]. The NF-κB family consists of five subunits in mammals: RelA (p65), RelB, c-Rel, p105/p50 (NF-κB1), and p100/p52 (NF-κB2), which can assemble into various homodimeric and heterodimeric complexes [29].

The p50–RelA dimer, one of the predominant activated NF-κB forms, has multifaceted functions in physiological and pathological situations. Sustained activation of this dimer under ischemic or chronic inflammatory conditions results in neuronal damage and tissue injury, while only transient activation is required for normal immune signaling and survival within the CNS [31].

NF-kB activation generally occurs along two pathways: the classic (canonical) and the alternative (non-canonical) one [32]. The canonical pathway activates p50, RelA, and c-rel, classical NF-kB subunits [33]. The alternative one deals with members of the NF-kB family retained by p100, namely p52 and RelB, which are considered non-classical subunits of NF-kB [34].

NF-kB activation leads to for the initialization and propagation of pro-inflammatory signals through pro-inflammatory mediators, such as TNF, IL-1, and lipopolysaccharide (LPS). This ensures that IκB undergoes phosphorylation and subsequent destruction to release p50 and p65 subunits to translocate into the nucleus and activate pro-inflammatory and survival genes [35]. This pathway plays a pivotal role in both early and delayed inflammatory reactions [36,37]. In addition, the non-canonical pathway is activated by receptors, such as CD40 and lymphotoxin-β, resulting in NF-κB-inducing kinase (NIK) and IκB kinase α (IKKα) activation. This results in converting p100 into p52, which, through association with RelB, governs genes associated primarily with lymphocyte development and B-cell maturation [34].

Alongside the NF-κB cascade, another crucial role in orchestrating the inflammatory response and progression of neurodegenerative diseases is played by the JAK/STAT pathway.

This pathway is a major cytokine-regulated signaling system and controls cell growth, apoptosis, and immune responses. Abnormal stimulation of the JAK/STAT pathway contributes to activation of myeloid cells and T lymphocytes, leading to neuronal damage and the progression of neuroinflammatory diseases [38].

STAT activation occurred through phosphorylation of serine and tyrosine residues in the carboxy-terminal domain. Phosphorylation of tyrosine induces dimerization of STAT, which is necessary for nuclear translocation from the cytoplasm, while serine phosphorylation enhances transcriptional activity [39]. Initially, STAT proteins were believed to be only cytoplasmic transcription factors that mediate signaling from cytokines and growth factors, but now it is clear that their regulation is very tightly tuned. One hand, phosphatases control the activity of STAT, while suppressor of cytokine signaling (SOCS) proteins modulate the activity of JAK. STATs are fundamental for processes including proliferation, differentiation, apoptosis, inflammation, and immune responses [40]. STAT1 appears to be especially important in immune regulation. IFN-mediated STAT1 signaling regulates immune cell growth, apoptosis, and cytokine production, particularly through Th1/Th2 response balance [41].

NF-κB and JAK/STAT coordinate the transcriptional control of inflammatory mediators, although other signaling pathways, such as those containing mitogen-activated protein kinases (MAPKs) are also involved in the stress and cytokine signaling cascades that modulate the precision of the inflammatory outcome.

The MAPK (Mitogen-Activated Protein Kinase) pathway is a conserved mechanism that regulates cellular response to stress and infection. In the central nervous system, it controls neuroinflammation through the activation of microglia and astrocytes [42]. The three main pathways (extracellular signal-regulated kinases, ERK; c-Jun N-terminal kinase, JNK; and p38 MAPK) have distinct functions: among these, p38 is closely linked to inflammation, stimulating the production of cytokines and ROS [43].

Another key role in the neuroinflammatory response is played by the NOD-like receptor protein 3 (NLRP3) inflammasome complex that serve as cytosolic sensor translating upstream signals into caspase-1 activation and cytokine maturation.

The NLRP3 inflammasome is a primary sensor of innate immunity that helps in distinguishing between invading pathogens and cellular stresses of endogenous origin. Activation of NLRP3 is mediated by stimuli including ion influx (potassium versus chloride), mitochondrial stress, release of reactive oxygen species and of mitochondrial DNA, and leakage of cathepsin B. Following activation, NLRP3 binds to NEK7, and they oligomerize to recruit ASC via PYD-PYD interactions. ASC subsequently recruits pro-caspase-1, yielding the NLRP3 inflammasome complex [44]. The consequence of this activation is the downstream activation of caspase-1, responsible for processing IL-1β and IL-18 and the cleavage of upon caspase-1 activation, gasdermin D (GSDMD). The GSDMD N-terminal fragment forms membrane pores leading to pyroptosis and the release of inflammatory cytokines. These cytokines enhance local inflammation and, through systemic circulation, also promote global inflammatory responses [45,46].

The activation of the NLRP3 inflammasome may provide benefits in the earlier stages of neuroinflammation by means of removal of pathogens and cellular debris.; However, sustained overactivation becomes detrimental, due to prolonged and excessive release of cytokines, perturbing CNS homeostasis and causing neuronal death [47].

Excessive activation of the NLRP3 inflammasome in glial cells also inhibits neuronal regeneration [48]. Pharmacological inhibition of NLRP3, by compounds as MCC950, has proven to be beneficial in various preclinical neurodegenerative disease models [49].

### 2.3. Epigenetic Regulation

Long noncoding RNAs (lncRNAs) are thought to be primary regulators in cellular processes, such as proliferation, apoptosis, and inflammation, and they are defined as transcripts longer than 200 nucleotides without coding potential [50]. lncRNAs display high expression levels in the central nervous system, probably reflecting the complexity of the brain, thereby requiring highly fine-tuned regulatory mechanisms for its development and function [51].

Mechanistically, they may exert their effects via post-transcriptional regulation or epigenetic modulation [52].

MicroRNAs (miRNAs) are minor noncoding RNAs (~20 nt) that bind to the 3′-UTR of messenger RNAs (mRNAs) to regulate around 30% of human genes [53]. They can also engage with lncRNAs to modulate their activities [54]. In the central nervous system, miRNAs play essential roles in the differentiation, survival, and regeneration of neurons [55], and their dysregulation may contribute to neurodegenerative disorders, [56]. The ceRNA (competing endogenous RNAs) hypothesis posits that mRNAs and lncRNAs are in competition for shared miRNAs, creating complex regulatory networks [57].

A summary of the main intracellular pathways mediating neuroinflammatory processes is provided in Table 1.

This table provides an overview of the main intracellular signaling pathways that are involved in neuroinflammation. It summarizes key molecules, principal biological functions, and the implications of each pathway in Alzheimer’s disease, with a particular focus on how these pathways promote chronic inflammation and neurodegeneration.

## 3. Neuroinflammation, Climate Change and Environment

Environmental stressors like pollutants, heat exposure, or dietary habits perturb the NF-κB, JAK/STAT, and NLRP3 inflammatory networks establishing a link between an externally mediated neuroinflammation with neuronal damage.

Recent epidemiological and experimental studies support the hypothesis that environmental factors cause neurodepolarization through activation of key neuroinflammatory pathways such as NF-kB, JAK/STAT, MAPK, and NLRP3.

Environmental stressors affecting neuroinflammation fall into three main categories: (1) air pollutants including particulate matter, ozone, sulfur dioxide, and nitrogen oxides; (2) climate-related stressors like exposure to heat waves and extreme temperatures; and (3) dietary and lifestyle factors.

Current studies suggest that air pollution may contribute to the development of neurodegenerative disorders by triggering neuroinflammatory processes and by altering white matter structure [58].

Impactful levels of air pollution have been shown to trigger the Aβ42 and α-synuclein accumulation in very early childhood, indicating that air pollution may influence early disease mechanisms and/or accelerate brain aging processes [59]. An alternate theory proposes that pollution-driven oxidative stress and nanoparticles may modify the rate of protein fibrillation and aggregation, influencing the soluble Aβ and α-synuclein. As such, pollution-induced changes in protein aggregation may constitute an early pathological event leading to neurodegenerative diseases [60].

The ‘multiple hit hypothesis’ has been proposed to suggest that environmental toxins can contribute to the pathogenesis of CNS disorders by producing different detrimental effects at different developmental stages in human life. Accumulating data from MRI studies in children exposed to heavy air pollution demonstrates structural brain alterations, particularly hyperintense white matter lesions in the prefrontal cortex, with possible associated cognitive deficits. Similarly, dogs exposed to the equivalent levels of air pollution display frontal lesions with evidence of vascular and endothelial damage and neuroinflammation [61]. Altogether, this suggests that young humans and animals alike may be particularly sensitive to air pollution’s inflammatory effects, with the potential for such effects to build cumulatively over the course of a lifetime.

While there is strong evidence for neuroinflammatory mechanisms due to air pollution, most mechanistic data comes from animal and in vitro studies. These studies often use exposure levels and durations that may not reflect real-life human conditions. Therefore, the translational validity of such studies is limited, and dose–response relationships in humans are only partially understood.

In epidemiological studies, confounding factors—including, but not limited to, socioeconomic status, comorbidities, access to healthcare, and concurrent exposures—may be skewing the associations between air pollutants and the observed cognitive decline. Even though many of these studies perform adjustments for multiple variables, one cannot exclude completely the possibility of residual confounding.

Despite the biological plausibility and the increasing agreement between experimental and observational studies, inferring a causal relation in humans requires longitudinal studies, rigorously controlled for exposure, with link to neuroinflammation and neurodegeneration established by the measurement of relevant biomarkers.

Above and beyond air pollutants, rising global temperatures represent another potent environmental trigger of neuroinflammation. Ozone is a highly reactive gas; its biological effects depend on its location within the atmosphere. Stratospheric ozone completely blocks UVB threats, while ground-level ozone is a harmful type [62]. Chronic exposure to ozone that resides in the troposphere causes oxidative stress and neuroinflammation, finally leading to the accumulation of proteins, neuronal loss, and cognitive deficits in exposed individuals [63,64]. Oxidative stress due to ozone exposure brings about mitochondrial dysfunction and activation of the NLRP3 inflammasome in glial cells. This cascade eventually leads to caspase-1 activation and subsequent secretion of IL-1β and IL-18, the two key mediators of the neuroinflammatory response described above. Animals exposed to ozone show impairments in memory and behavior, neurogenesis, and synaptic structures, in a manner that could be similar in patients with Alzheimer’s disease [65]. Nevertheless, these effects have been mainly documented in controlled laboratory conditions, with uncertainty about their magnitudes and clinical relevance in people under usual environmental exposure.

Beyond ozone, other combustion-related gases may play a role in neuroinflammation. Among these, sulfur dioxide (SO_2_) and nitrogen oxides (NO_x_) have been extensively linked to cognitive decline and neuronal injury.

SO_2_ is a byproduct of burning sulfur-containing fuels as well as derived through industrial activities, which causes serious respiratory, cardiopulmonary and neurodegenerative disorders [66]. Exposure to SO_2_ entails neuroinflammation-related problems, synaptic injury, and most importantly, learning and memory deficits in animals. Further, epidemiological evidence indicates that elevated concentrations of SO_2_ hasten the speed of cognitive decline in Alzheimer’s disease, inhibit specific cognitive functions, and may increase the risk of developing multiple sclerosis at a younger age [67,68]. These gaseous pollutants produce systemic oxidative stress and damage the integrity of the vasculature, allowing entry of peripheral immune cells into the bloodstream through the blood–brain barrier thereby stimulating microglial NF-κB activation and JAK/STAT signaling. Even in this case, establishing a causal relation is difficult because SO_2_ exposure may occur simultaneously with other air pollutants and other determinants, e. g., socio-economic.

NOx, and, in particular, nitrogen dioxide (NO_2_), can deposit deeply in the lungs possibly causing exposure to the agents for a long time [69]. Animal studies have shown that exposure to NO_2_ leads to memory impairment, increased accumulation of Aβ42, and Alzheimer’s-like pathology, in addition to effects on vascular dementia and synaptic plasticity. Likewise, population studies have shown that higher levels of NO_2_ correlate with higher risks of dementia and Parkinson’s disease and are related to greater thinning of the cortex in affected brain regions and likely increased vulnerability of the developing cerebral cortex [70,71]. Nitrogen oxides elevate the permeability of the BBB and activate microglial priming via TLR4-dependent activation of NF-κB, leading to sustained release of TNF-α and IL-6.

All these pollutants share common pathways leading to oxidative stress, mitochondrial dysfunction, and initiation of cascades of inflammatory signaling like NF-κB and MAPK. These molecular events connect the environmental exposure with neuronal vulnerability.

Biomolecular sequences undergo epigenetic alterations, such as DNA methylation, changes in histone modifications, or any dysregulation of microRNA. Exposure to PM, NO_2_, and O_3_ can induce epigenetic alterations that regulate the expression of genes responsible for oxidative stress response, inflammation, synaptic plasticity, and neuronal survival [72].

Air pollution can act at the molecular level, increasing ROS generation that produces oxidative damage to DNA, lipids, and proteins, which activates stress-response signaling pathways, such as transcription factor NF-κB, MAPK, and JNK toward the promotion of neuroinflammation [73]. Pollutants also alter the balance of prostaglandin production through COX-2, thereby affecting neuronal signaling and leading to cognitive deficits; dysregulation of microRNA gets involved with the modulation of genes involved in apoptosis, synaptic function, and neuroinflammatory responses, whereas the process of histone acetylation and methylation changes can either repress neuroprotective genes or upregulate pro-inflammatory ones [74]. Newly internalized particles can cause mitochondrial dysfunction along with NLRP3 inflammasome activation in microglia, simulating the intracellular inflammatory cascades induced by classical air pollutants.

In AD, dysregulated gene expression may favor Aβ accumulation and tau hyperphosphorylation, while in PD cases, disrupted mitochondrial integrity and oxidative stress place dopaminergic neurons in a particularly vulnerable status. Some of these epigenetic changes may even be inherited and would likely render future generations more susceptible to neurodegeneration [75].

Among the novel environmental pollutants, micro- and nanoplastics (MPs and NPs) have proved to be critically neurotoxic, since they can be inhaled, and then cross the blood–brain barrier, inducing oxidative stress (by increasing ROS levels and lipid peroxidation, and decreasing antioxidants), neuroinflammation, and neuronal damage, with their accumulation confirmed in both animal and human brains [76,77]. Moreover, the accumulation of airborne MPs/NPs disrupts the normal neurotransmission [77]. It is noteworthy that the majority of presently available data has been obtained from animal and cell models, whereas the extent of brain accumulation and toxicity in humans is still being investigated. As observed by Lee et al., the oral administration of polystyrene (PS)-MPs to mice for eight weeks had negative consequences especially on the hippocampus with microglial activation, inflammation, downregulation of several genes involved in synaptic plasticity and memory, and upregulation of synaptic glutamate AMPA (α-amino-3-hydroxy-5-methyl-4-isoxazolepropionic acid) receptors. Interestingly, since the ablation of the vagus nerve improved memory in the exposed mice, the authors hypothesized a vagal-dependent neurotoxicity due to PS-MPs [78]. Other pathways like ErbB4 activation, may counteract these toxic effects and provide some therapeutic options [79,80,81].

Cumulatively, this evidence highlights how microplastics represent a new environmental factor capable of fueling neuroinflammation and neurodegenerative vulnerability.

Besides chemical pollutants, climate-related physical stressors, especially heat and temperature extremes, are important drivers of neuroinflammation.

A great deal of public concern remains about brain health due to the heavy impact from global warming and heatwaves on neuroinflammation. Heat stress stimulates glial cells to release IL-1β, IL-6, and TNF-α. These pro-inflammatory cytokines suppress neurogenesis in the hippocampus and may activate caspases for apoptosis and mitochondrial dysfunction. In animal models, heat stress causes excitotoxicity due to excess excitatory neurotransmitters, like glutamate and aspartate, and decreased inhibitory neurotransmitters, such as GABA and glycine. In the process excitotoxicity, intracellular calcium rises, stimulating N-Methyl-D-Aspartate receptor (NMDAr), which is followed by oxidative stress, DNA damage, and apoptosis [82,83]. In astrocytes and microglia, heat exposure at the molecular level stimulates MAPK and NF-κB signaling, thereby enhancing cytokine production and mimicking pollutant-induced inflammatory cascades. Transcriptomic analyses on mice subjected to several heat waves show upregulated inflammatory pathways in both systemic and synaptic networks, with concomitant increased expression of related molecules, such as S100B, IL-17, NOD1/2, and Triggering Receptor Expressed on Myeloid Cells 1 (TREM1), and downregulation of protective pathways, such as aryl hydrocarbon receptor (AhR) signaling [84]. This suggests that thermal stress may affect the body locally but also involve extrinsic effectors, such as the liver-brain axis in enhancing neuroinflammatory cascades.

Ongoing research is investigating therapeutic targets for heat-induced neuroinflammation. For example, treatment with β-hydroxylbutiryate in the form of intraperitoneal injection reduced hippocampal neuroinflammation in heat-stressed mice an effect paralleled by a decreased level of cytokines and regulated microglial activation [85]. Collectively, these findings implicate climate change and extreme heat exposure as powerful environmental triggers of neuroinflammation capable of accelerating neurodegenerative processes and increasing the vulnerability of predisposed individuals.

Indoor PM and VOCs elicit similar oxidative and inflammatory effects, leading to structural brain changes [86]. Using many different molecular mechanisms, fine particulates and transition metals generate ROS that activate the microglial and astrocytic NF-κB and MAPK signaling pathways, ultimately inducing the transcription of pro-inflammatory cytokines and promoting the sustenance of chronic glial activation.

Finally, dietary and metabolic factors constitute a modifiable category of environmental stressors that can interact with pollutants and thermal stress to exacerbate neuroinflammatory pathways. Finally, dietary and metabolic factors represent a modifiable class of environmental stressors that synergize with pollutants and heat stress to amplify neuroinflammatory processes. In the last few decades, the link between dietary habits and neuroinflammation has been increasingly elucidated. Current evidence suggests that high-fat (HF) diets induce chronic, low-grade inflammation in the brain, which may be responsible for cognitive impairment [87]. HF diet-fed mice exhibited higher brain lipid peroxidation, and oxidative damage, compared to normal fed mice [88]. The brain mitochondrial disfunction observed in these conditions, highlighted by increased ROS production, changes in mitochondrial membrane potential, and opening of mitochondrial permeability transition pores (mPTP), leads to neuroinflammation through different mechanisms. On one hand, damaged mitochondria represent an important source of damage-associated molecular patterns (DAMPS), stimulating innate immune pathways (e.g., the activation of NLRP3 inflammasome and the macrophage production of IL-1β).

On the other hand, brain inflammatory cells (e.g., microglia) undergo a metabolic remodeling, promoting the release of pro-inflammatory mediators [87]. The Western diet (WD), rich in saturated fatty acids (SFAs) and refined carbohydrates, represents a challenge for the gut–brain axis. The WD causes gut microbiota (GM) dysbiosis, an increased bacterial production of LPS translocating into the bloodstream through the “leaky gut”, and the activation of Toll-like receptor 4 (TLR4) on gut macrophages, promoting the NF-kB signaling pathway, and the release of inflammatory cytokines involved in local and systemic inflammation. In addition, SFAs passing through the BBB bind to TLRs located on microglia, inducing neuroinflammation in a similar manner [89,90]. Saturated fatty acids can cross the BBB and engage TLR4 receptors on microglia, activating the NF-κB and ROS pathways. This is suggestive of an initial inflammatory signaling pathway, like that found for environmental pollutants, and perhaps implies a shared mechanism [91].

A conceptual framework integrating environmental stressors, inflammatory signaling, and neurodegeneration is shown in Figure 1.

## 4. Neuroinflammation in Alzheimer’s Disease

The pathogenesis of AD is a classic example of chronic neuroinflammation. In AD, inflammation is an active and contributory process involving microglial activation, cytokine release, and immune-metabolic dysregulation rather than a consequence of a secondary event arising from amyloid-β and tau pathology.

AD encompasses both positive and negative lesions within the CNS. Positive lesions refer to the accumulation of neurofibrillary tangles (NFTs) in neurons, deriving from hyperphosphorylated tau protein, and extracellular plaques, consisting of β-amyloid (Aβ) peptide. Negative lesions result from the brain atrophy due to neural and synaptic loss [92].

Neuroinflammation is the main driver of Alzheimer’s disease, not simply a consequence of the disorder. Sustained microglial activation, once considered adaptive to Aβ, is now viewed as maladaptive, worsening amyloid toxicity and exacerbating tau pathology [93].

A chronic inflammatory role in AD is supported by the elevation of inflammatory biomarkers in blood and cerebrospinal fluid (CSF) studies, as well as by positron emission tomography (PET) imaging with translocator protein (TSPO) ligands, which detect microglial activation in vivo. However, classical markers, such as CRP and other pro-inflammatory cytokines, have repeatedly shown inconsistent results, hampering their diagnostic reliability in AD neuroinflammation [94].

The role of genetics in AD is supported by various genome-wide association studies (GWAS). Such studies established direct genetic correlation relating inflammatory conditions to disease mechanisms. A twin study has estimated that the genetic inheritance accounts for 56–79% AD risk [95]. Over the past 15 years, with GWAS and next-generation sequencing, more than 80 independent loci have been associated with an increased risk for developing AD [96]. From pathway analysis of genome-wide association studies, it can be inferred that dysregulation of normal adaptive immune responses is one of many factors that contribute to the development of AD. Most common variants enhancing the risk of AD via enhanced activation of regulatory elements in microglia; significantly, one-quarter of all immune-related risk genes identified are strongly or exclusively expressed in microglia [97].

Genome-wide association studies have provided support for a strong immunological role in the susceptibility of AD, as many of the most significant loci act in the control of microglial and related innate immune functions. TREM2 encodes a receptor that is essential for phagocytosis and sensing of lipids by microglia. Loss-of-function variants reduce microglial clearance of amyloid-β and apoptotic debris, thus causing persistent NF-κB activation and cytokine release [98]. CD33, another microglial receptor, inhibits phagocytosis using sialic acid–dependent signaling; at-risk alleles enhance its inhibitory effects and reduce the uptake of Aβ while promoting chronic glial activation [99].

CR1 and CLU modulate the complement cascade, affecting synaptic pruning and immune clearance, while aberrant regulation of these proteins leads to excessive complement activation and neuronal loss [100,101]. Similarly, variation in ABCA7 function may affect lipid efflux and vesicular transport in microglia, thereby compromising Aβ degradation and promoting NLRP3 inflammasome activation [102]. Subsequent altered lipid handling increases inflammatory signaling-promoting and tau-phosphorylating effects of the major apolipoprotein, APOE, particularly noticeable or its ε4 allele associated with altered microglial reactivity and cholesterol transport.

Cumulatively, these genes define a molecular mechanism through which defective immune regulation and poor clearance of waste products give rise to prolonged activation of microglial cells and cytokine release in relation to genetic vulnerability to neuroinflammation and the amyloid/tau cascades of Alzheimer’s neuropathology. This is further corroborated by the meta-GWAS data recently issued from the European Alzheimer & Dementia Biobank, whereby most immune loci were reconfirmed, and several new variants were identified in the three genes SHARPIN, RBCK1, and OTULIN, which are critical components of the linear ubiquitin chain assembly complex (LUBAC). LUBAC is of fundamental importance for the activation of the NLRP3 inflammasome and autophagy, linking ubiquitin signaling to the dysregulation of immune responses and the clearance of misfolded proteins, including the TDP-43 inclusions [103].

Along with this, the same study highlighted the important role of the TNF signaling pathway as it uncovered loci such as ADAM17 (the most important metalloproteinase activated by TNF) and TNIP1 (an inhibitor of TNF signaling) [104]. This included other relevant genes, such as SPPL2A, which participates in non-canonical TNF shedding, or progranulin (PGRN/GRN), involved in the dual role of the TNF receptor ligand and antagonist [105].

Beyond innate, genetic studies also imply that adaptive immune mechanisms may be linked to increased risk of developing AD. This regards especially the involvement of the HLA-DRB1 subset, with the HLA-DRB1*04 subtype appearing protective perhaps through targeting tau by the immune system, particularly its K311-acetylated form, which facilitates the aggregation of tau. Tau pathology depends on neuroinflammation induced by Aβ42, with microglia serving as a link between these two major pathological features of AD [106,107]. Tau pathology also interacts with microglial activation, further amplifying neuroinflammatory damage [108,109]. The current work in functional genomics will combine these genetic discoveries into innovative therapeutic approaches.

An overview of the interplay between Aβ, tau pathology, and neuroinflammatory responses, including their downstream effects on neuronal survival, is summarized in Table 2.

The contribution to AD pathogenesis by some of these cell types is incomplete. For example, the activation of astrocytes in AD is controversial. On one hand, activated astrocytes have been shown to clear protein aggregates locally in the brain [111]. On the other hand, large groups of pro-inflammatory astrocytes can be found in postmortem brain tissue from patients with AD, indicating a possible role in neurodegeneration [110].

Oligodendrocytes are vital in the pathogenesis of Alzheimer’s disease. Cholesterol homeostasis is impaired by APOE4, hindering myelination, whereas Erk1/2 signaling favors myelin repair while attenuating Aβ-associated pathology. Key players in myelination/axonal support are myelin protein genes (Mbp, Mobp, Olig2, Mag, Mog) and new markers (PLP1, ST18) [112,113]. BACE1, AK5, and PIP4K2A dysregulation affects production of Aβ, triggering neuroinflammation, apoptosis, and energy metabolism. Regulatory modules involving APOE and CLU further indicate oligodendrocytes’ importance in AD [114,115].

GFAP-positive astrocytes have important roles in AD, such as regulating neuroinflammation, maintaining the blood–brain barrier, and supporting glutamate homeostasis. These astrocytes can adopt either a neurotoxic (A1) or neuroprotective (A2) state. By targeting pathways like STAT3, NF-κB, or RAGE—through NGFR, AQP4, or short-chain fatty acids—it may be possible to boost protective effects, reduce Aβ and tau pathology, prevent cognitive decline, and enhance neuroprotection [116,117].

As main immune actors of the CNS, microglia, astrocytes, and oligodendrocytes are always investigated. However, Alzheimer’s disease is increasingly considered a state in which also peripheral immunity is actively involved. Aging and chronic inflammation can compromise the blood–brain barrier (BBB), allowing soluble mediators and peripheral immune cells to interact with neural tissue. Circulating cytokines augment glial activation, while lymphocytes entering the CNS are probably directly involved in neuronal injury and amyloid clearance. In this context, B cells, T cells, and natural killer (NK) cells become additional elements linking systemic immune dysregulation to central neuroinflammatory processes [118,119].

In AD, transcriptomic B-cell changes were observed. Differential gene expression analysis revealed that KIR3DL2, OPCT, and PPP2R2B were upregulated, while FRAT2, WWC3, and SPG20 were downregulated, all implicated in neurodegeneration [120]. Single-cell RNA sequencing (scRNA-seq) further gave rise to the identification of a novel B-cell phenotype characterized by high CD45 expression, enhanced phagocytosis and chemotaxis, and secretion of multiple chemokines that recruit peripheral immune cells via the CCL signaling pathway. This change could arise from the increased expression of myeloid-associated transcription factors, such as the CEBP family, and reduced expression of lymphoid transcription factors like Pax5 [121].

T cells are equally seen as important contributors to AD pathology. The CXCL10-CXCR3 axis mediates T-cell infiltration and neuronal injury, whereas CD8^+^ T-cell infiltration activates microglia, aggravating neuroinflammation and neurodegeneration [122]. Cis-regulatory elements co-accessible with the CXCR3 promoter in peripheral CD8^+^ T-cells suggest an epigenetic mechanism that may impart susceptibility to AD [123].

NK cells are also involved in the pathogenesis of AD. Their study revealed 17 AD-associated marker genes in NK cells, including EEF1B2, GPR56, H3-3B, and ZEB2 that might influence infiltration of immune cells [124]. Cell communication analysis revealed NK cell subsets with characteristic signature genes, including RPLP2, RPSA, and RPL18A. One such subgroup characterized by upregulation of CX3CR1, TBX21, MYOM2, DUSP1, and ZFP36L2 was found to be negatively correlated with cognitive function in AD patients [125]. NK cells engage with other immune cells, specifically dendritic cells and macrophages, thereby shaping the immune landscape. Upon activation, NK cells can mature or kill dendritic cells, therefore indirectly modulating T cell priming [126]. In addition, NK cells within the periphery can penetrate the brain and, through the STAT3 signaling pathway, regulate the transcription of immune response genes, thereby amplifying the neuroinflammation [127].

Recent studies indicate that NLRP3 inflammasome activation in microglia directly contributes to AD pathogenesis. IL-1β/IL-18 maturation and subsequent pyroptosis promote synaptic loss, amplify Aβ aggregation, and induce neuronal degeneration. Pharmacological inhibition of NLRP3 reduces inflammation and pathology in preclinical models, suggesting a promising therapeutic target.

Acosta-Martínez et al. conducted research into neuroinflammatory mechanisms in Alzheimer’s disease. Their study points to the fact that the inflammatory response is not generalized but shows variation across different brain regions. Results that support this notion imply neuroinflammation could differentially contribute to AD pathogenesis depending on the regional brain vulnerability aspect, which could provide cues for designing more targeted therapeutic strategies [91].

Mutually enhancing genetic and environmental factors demonstrate their role in increasing the risk for AD. Of all the genes implicated in sporadic AD, the most potent remains the APOE ε4 allele, which seems to intensify the neurotoxic effects of environmental stressors such as air pollution, high-fat diet, and heat exposure. Epidemiological and experimental studies showed that in comparison to non-APOE ε4 carriers, APOE ε4 carriers show increased microglial activation, blood–brain barrier disruption, and deposition of amyloid-β following exposure to particulate matter or ozone [95,128,129]. On a molecular level, APOE ε4 modifies lipid metabolism and hinders Aβ clearance, thereby sensitizing neurons to oxidative and inflammatory damage induced by pollutants and metabolic stress [98,102,130]. Likewise, polymorphisms in TREM2 and CD33 that affect phagocytic efficacy in microglia may counteract the effect of environmental stressors on the neuroinflammatory response, turning it into a chronic rather than an adaptive response [97,99].

Thus, environmental stressors catalyze the manifestation of gene vulnerability and accelerate the neurodegenerative process under common conditions of oxidative stress, mitochondrial dysfunction, and immune dysregulation. It derives that understanding this interaction is crucial for prevention strategies in patients with environmentally sensitive genotypes such as APOE ε4.

## 5. Alzheimer’s Disease and Environment

The relationship between environmental stressors and Alzheimer’s disease may be conceptually categorized into three major categories of risk factors: (1) air pollution and toxins in nature, such as particulate matter, heavy metals, and volatile organic compounds, that induce oxidative stress and neuroinflammatory activation; (2) lifestyle and dietary factors like the western-type diet (high-fat diet), lack of physical exercise, and metabolic disturbances that modulate systemic inflammation and amyloid metabolism; and (3) climate-related physical stressors, including heat exposure and temperature changes, which alter neuronal and glial homeostasis through stress-related signaling pathways.

Dietary exposures represent one of the most accessible and modifiable environmental factors influencing neurodegenerative risk. Several studies investigated the association between dietary patterns and cognitive changes. For instance, in an Australian study of aging, Gardener et al. detected better executive functions after 3 years of Mediterranean diet (MD) in apolipoprotein E (APOE) ɛ4 allele carriers, whereas the Western diet was associated with a higher cognitive impairment in APOE ɛ4 allele non-carriers [131]. In another study, HF-diet fed mice exhibited hyperphosphorylation at serine residues of AD-associated tau protein in the hippocampus, linked to neurofibrillary pathology. This effect was prevented by the concomitant dietary administration of capsaicin, a transient receptor potential vanilloid 1 (TRPV1) agonist reducing insulin resistance, which restored the phosphatidylinositol 3 kinase/protein kinase B (PI3K/AKT) signaling pathway [132]. Similarly, an increased tau phosphorylation was observed in the hippocampus of rats after only a week of high-fat-and-fructose diet, which also induced reactive astrocyte activation, and reduced dendritic arborization in CA1 neurons, as well as the amounts of synaptophysin at this level [133]. Similarly, a high-fat, high-sucrose WD in rats significantly reduced long-term potentiation (LTP), an electrophysiologic marker of synaptic plasticity, at Schaffer collateral-CA1 synapses. Researchers also found an altered brain signaling in WD-fed mice with lower levels of brain-derived neurotrophic factor (BDNF) and postsynaptic density protein 95 (PSD-95), linked to postsynaptic degeneration, whereas ERK1/2 was upregulated [134]. Nowadays, there is evidence that the adherence to healthier dietary approaches, such as MD, rich in fibers, polyunsaturated fats, and neuroprotective compounds (e.g., polyphenols and vitamins) can prevent or decelerate cognitive decline, improving brain circulation, lowering neuroinflammation and oxidative stress, and ameliorating both neurogenesis and neuroplasticity [135]. Braden-Kuhle et al. have recently evaluated the effects on mice of two tailored diets, respectively, reflecting the composition of human MD and typical American diet (TAD). MD mice exhibited enhanced spatial memory and exploratory behavior, lower levels of serum TNF-α and central Aβ, reduced neuroinflammation and BDNF loss, as compared to TAD mice, together with a metabolic improvement in terms of body weight and abdominal fat [136]. In recent years, randomized controlled trials (RCTs) have evaluated the effects of diet on AD biomarkers, brain perfusion, and cognitive function. For instance, in patients with mild cognitive impairment, Hoscheidt et al. reported increased Aβ42/40 ratios in cerebrospinal fluid following 4 weeks of WD, and decreased levels after MD. They also found that MD increased and WD decreased cerebral perfusion in participants with normal cognition [137]. However, a trial examining the effects on brain health of a revised MD showed no significant impact on cognition and radiological outcomes in older individuals without cognitive impairment. It is important to highlight that this kind of studies present several limitations (e.g., socioeconomic background and education), which may compromise the interpretation of the results and make them not applicable to general population [138].

Beyond nutrition, climatic variables such as temperature extremes and air quality also shape neurodegenerative vulnerability. Climate change influences the incidence and the burden of AD and related dementias (ADRD) both directly and indirectly. On the one hand, extremely high temperatures, flooding and environmental pollution directly compromise brain health, on the other hand, they may limit health behaviors (e.g., physical activity and restful sleep), care access and quality, and cause socioeconomic inequalities [139,140]. In a large population-based cohort study, Delaney et al. noticed that extreme heat is particularly harmful for older people with ADRD, increasing the risk for hospitalization. It was estimated that each day of extreme heat contributed to approximately 5360 added hospital admissions of patients with ADRD across the United States [141]. Likewise, the association between higher warm season-temperatures and emergency department visits for ADRD was observed in 5 US states, further suggesting the vulnerability of these patients to ambient temperature [142]. In China, an increase in mortality due to ADRD among the elderly was observed during and after heatwave events between 2013 and 2020 [143].

At a molecular point of view, heat-related oxidative stress in one of the most important mechanisms contributing to neuronal damage and AD exacerbation [82]. As previously described by the mitochondrial cascade hypothesis, mitochondrial disfunction and ROS production trigger the expression of amyloid precursor protein (APP) and the accumulation of Aβ in sporadic, late-onset AD [144]. Aggregated Aβ and neurofibrillary tangles stimulate in turn the release of ROS, ultimately leading to protein, lipid and DNA damage, and consequently neuronal death [145].

Epigenetic changes such as DNA methylation and miRNA dysregulation may mediate environmental effects on AD risk [146,147]. Among these small noncoding RNAs, miR-125b induces glycogen synthase kinase-3β (GSK3β) and tau phosphorylation by targeting NCAM (neural cell adhesion molecule), involved in the AD neuropathological progression [148].

Recent epidemiological studies have highlighted a positive correlation between air pollutants and ADRD, which is corroborated by findings from animal studies. Most studies reported increased brain levels of Aβ/NFT, tau hyperphosphorylation, astrogliosis, microgliosis, and neuronal cell loss, following a prolonged exposure to PM and O_3_ [129]. Air pollutants enhance ADRD phenotypes via inflammatory mechanisms both directly, by crossing the BBB from systemic circulation and activating glial cells, and indirectly, by inducing pulmonary and systemic inflammation, and the activation of the lung–brain axis [129].

In a study conducted on mice and nematodes, Haghani et al. examined the connection between gestational exposure to PM and the risk of cognitive decline in later life stages. They noticed DNA damage and oxidative stress in mice at postnatal day 5, with an upregulation of AD-related genes, such as APP, Psen1 (presenilin-1) and tau. Similarly, they found an early upregulation of inflammatory genes and AD-gene homologs in nematodes, resulting in an accelerated paralysis phenotype in adults, linked to Aβ42 production [149]. A longitudinal study about the effects of air pollution on 269 individuals with mild cognitive decline or initial dementia reported a faster memory loss after chronic exposure to sulfur dioxide (SO_2_), and a faster decline in visuospatial abilities after being chronically exposed to PM_2.5_ [150]. In an autoptic study conducted on 602 individuals with dementia and/or movement disorders, PM_2.5_ exposure before death, assessed through a spatiotemporal prediction model, was associated with greater cognitive impairment and neuropathologic changes [151]. Additionally, a recent systematic review with a Burden of Proof meta-analysis about the effects of long-term exposure to PM_2.5_ on dementia reported a significant association between fine PM and AD (three stars in a 1–5-star rating scale from weak to very strong evidence) [152].

Pesticides induce oxidative stress and mitochondrial dysfunction, accelerating Aβ and tau pathology [153]. Increased concentrations and abnormal distributions of metals are implicated in the pathogenesis of AD. Early on, some heavy metals—such as arsenic, cadmium, lead, and mercury—were shown to cross the blood–brain barrier. These metals accumulate in the CNS, eventually resulting in neurodegeneration. Aluminum is found to be elevated in AD brains. It co-localizes with amyloid plaques and provokes AD indications like oxidative stress, calcium dysregulation, and cholinergic damage. Mercury is also linked with cognitive impairment and dementia. Exposure to arsenic from the environment leads to cognitive decline. It contributes to AD-related mortality by triggering redox imbalance, mitochondrial dysfunction, vascular compromise, and loss of neuronal survival [154,155]. Interestingly, in a quantitative meta-analysis, the circulatory levels of toxic metals, such as aluminum, mercury, and cadmium, were found to be significantly higher in AD patients, compared to controls, suggesting a possible role in AD [156].

A comparative summary of the available preclinical, clinical, and epidemiological evidence for each environmental stressor is provided in Table 3.

## 6. Conclusions and Future Perspectives

The evidence presented in this review points to a theory that may link neuroinflammation with neurodegeneration by both intrinsic (genetic and metabolic) and extrinsic (environmental and lifestyle) ways [157,158]. Such crosstalk between the brain and the periphery translates external insults into cellular dysfunction and progressive cognitive decline, acting via activation of microglia, cytokine signaling, oxidative stress, and immune infiltration. This interconnected understanding of AD shifts in part the emphasis away from the isolated pathogenic hypothesis toward a more systems-level model, wherein inflammation constitutes the trigger for the initiation and the amplifier for AD progression [159].

Based on this integrative perspective, environmental exposures are rapidly emerging among the leading global health challenges of our time. Cumulatively, epidemiological, experimental, and mechanistic data underscore the impact of environment and climate change on neurological health. The main mechanisms through which environmental factors influence the neuroinflammation-neurodegeneration sequence have been discussed introducing the novel concept of an environment-neuroinflammation-neurodegeneration, especially for AD. Presently this remains simply an hypothesis because, available studies there are predominantly observational in nature, with several confounding factors (e.g., lifestyle, and socioeconomic factors), and an imperfect exposure assessment (e.g., self-reported data, and different exposure metrics across studies).

Future research needs to investigate and distinguish the mechanistic hierarchies of neuroinflammatory processes connecting environmental risk factors and neurodegeneration. Cutting-edge methodologies such as single-cell multi-omics, spatial transcriptomics, and metabolomic profiling could allow precise mapping of microglial and astrocytic activation states in relation to specific environmental factors. These data could be subsequently integrated into computational models for dynamic reconstruction of immune-neuronal networks, eventually identifying the causal nodes responsible for the failure of transition from adaptive to chronic inflammation.

Simultaneously, longitudinal studies in humans conducted in parallel with such complementary peripheral and central biomarkers will indicate when and how immune changes precede neurodegeneration-their identification being a means to define windows of opportunity for an early preventive intervention. Ultimately, this conceptual and systems-level orientation would underpin precision immunomodulatory treatment of Alzheimer’s disease [157,160,161].

Longitudinal studies with extended follow-up would help assess the long-term neurological consequences of exposure to climate-related stressors. These studies should account for risk modifiers, behavioral factors, and comorbidities, and incorporate larger populations across different ages, ethnic and socioeconomic conditions, to provide findings that better represent the global population [162].

A crucial focus should be the identification of early biomarkers of climate-related neurological damage, based on neuroimaging, electrophysiology, cerebrospinal fluid, and blood biomarkers in people exposed to such stressors. Establishing sensitive and specific biomarkers could significantly increase the early diagnostic capabilities, and allow the risk stratification, as well as the assessment of preventive strategies [163].

The most likely candidates for biomarkers for environmental neurodegenerative damage seem to be (1) inflammatory and immune mediators such as IL-1β, IL-6, TNF-α, and chemokines, which reflect systemic and glial activation [164,165]; (2) oxidative stress and mitochondrial dysfunction markers, including 4-hydroxynonenal (4-HNE), ROS, and mitochondrial DNA damage, which increase upon exposure to pollutants and thermal stress [166]; (3) inflammasome components, such as NLRP3, caspase-1, and IL-18, providing direct mechanistic links between environmental stressors and neuroinflammation [167]; (4) traditional Alzheimer’s biomarkers (Aβ42/40 ratio, phosphorylated tau), capturing the downstream effects of environmental insults onto amyloid and tau pathology [137]; (5) epigenetic and transcriptomic markers, including DNA methylation profiles and microRNAs (miR-125b, miR-34a, and miR-146a), which are sensitive to air pollution and heavy metal exposure and regulate inflammatory and neurodegenerative pathways [168]; and (6) advanced neuroimaging indicators, particularly TSPO-PET ligands, which non-invasively detect microglial activation and neuroinflammation related to environmental exposures [169].

Integrating these different biomarkers offers the best chance to detect early environment-driven neurological changes. especially before clinical decline begins.

In the last few years, novel therapeutic approaches have been proposed to face climate-related neuroinflammation and neurodegeneration. The central role of neuroinflammation in AD pathogenesis has encouraged scientific community to develop therapeutic strategies inhibiting microglia activation (e.g., microglial receptor inhibitors), inflammasome pathway, and cytokine cascades [170]. Interestingly, an emerging field of research is the signaling of leptin, an anorexigenic hormone produced by adipocytes and involved in energy homeostasis. PM_2.5_, containing LPS, activates TLR4 pathway and increases the levels of the suppressor of cytokine signaling 3 (SOCS3), inhibiting the leptin action. Since leptin takes part in neuroprotection, preventing Aβ accumulation and tau hyperphosphorylation, targeting the TLR4-SOCS3-leptin axis may ameliorate the metabolic and inflammatory imbalance caused by pollutants [171]. A useful antioxidant function can be provided by N-acetyl cysteine (NAC), a Food and Drug Administration (FDA)-approved agent. Ontawong and colleagues reported the ability of NAC to reduce the brain levels of Aβ40 and 4-hydroxynonenal(4-HNE), an oxidative stress marker, and BBB leakage in 5xFAD (five familial AD mutations) mice, compared to wild-type mice, thus improving their cognition [165]. In recent years, NLRP3 inhibition has emerged as a promising therapeutic approach in preclinical studies. For example, small molecules like MCC950 and dapansutrile (OLT1177) have ameliorated cognition in several mouse models [172]. However, the clinical translation of these data remains weak, since many inhibitors show peripheral rather than central activity with an uncertain BBB penetration, and the long-term effects of a chronic inflammasome suppression on the host immune system have not been evaluated yet. To address these issues, novel brain-penetrant NLRP3 inhibitors and MCC950-derived molecules are being developed at early stages [167,173,174].

In addition, dietary interventions aimed at reducing neuroinflammation have been largely proposed. For instance, polyphenols, phytonutrients provided by plant-based foods, exhibit antioxidant, anti-inflammatory, and anti-amyloid properties. Their role in neuronal viability, synaptic plasticity, and cognitive function is mediated by their ability to modulate crucial signaling pathways, such as AKT, Nuclear factor erythroid 2-related factor 2 (Nrf2), STAT, and MAPK, involved in neuroprotection [175]. Among phenolic compounds, resveratrol, which is present in berries, peanuts, and the skin of red grapes, induced antioxidant enzymes (e.g., catalase and superoxide dismutase 2), and inhibited the inflammasome pathway in microglia treated with the pro-inflammatory monomeric C-reactive protein (mCRP) in vitro [176]. Resveratrol also inhibited the oligomeric Aβ-induced microglial activation via Nicotinamide Adenine Dinucleotide Phosphate (NADPH) oxidase, thus suggesting its therapeutic potential in AD [177]. Another polyphenol compound, curcumin, may contribute to cognitive improvement in neurodegenerative disorders. Animal studies underline its role in inhibiting Aβ aggregation, reducing oxidative stress, and modulating glial cells [178]. In humans, multidomain lifestyle interventions, including nutritional guidance, physical exercise, and social stimulation, have shown modest cognitive benefits, particularly in older adults, as well as important cardiometabolic co-benefits [179,180]. It is recommended to develop larger RCTs to explore dietary interventions and their correlation with objective biomarkers (e.g., blood and CSF inflammatory panels, or neuroimaging) and address adherence barriers, such as food access and education.

In the future, epigenetic modulation and gene-targeting therapy will hopefully be innovative approaches to modify neuroinflammation in neurodegenerative diseases, providing neuroprotection and disease-modifying benefits [129,181].

An emerging topic of research is the mental health impact of environmental stressors, commonly referred to as “ecological anxiety” or “ecological grief”. In times of growing concerns about climate and pollution, it is essential that researchers, clinicians, public health practitioners, and decision-makers work together to promote strategies to support, strengthen, and heal [182].

On a wider scale, public health interventions are needed to minimize the exposure to pollutants and their impact on human health. For instance, as observed in Beijing, promoting air filtration in areas with extremely high outdoor pollution may reduce the indoor pollutant levels with potential health benefits (e.g., reduced systemic inflammation) [183]. Similarly, a recent double-blinded RCT in Hong Kong underlines the positive effects on cardiovascular health of long-term indoor air purifiers [184]. Moreover, recent policy interventions, such as London’s ULEZ (Ultra Low Emission Zone) expansion with stricter regulations applied to most vehicles, have demonstrated significant improvement in air quality [185]. As described in a German qualitative study, co-designed urban planning involving community members can foster brain health and prevent dementia. Three main themes emerged: social inclusion, accessibility to services, and local recreation/well-being (e.g., safety, urban greenery, and heat mitigation) [186].

Overall, expanding urban green areas, using air purifiers, enforcing strict emission controls, enhancing mitigation strategies, and promoting public awareness campaigns can help address this issue from multiple angles, in order to safeguard brain health for future generations [187].

## Figures and Tables

**Figure 1 cimb-47-00959-f001:**
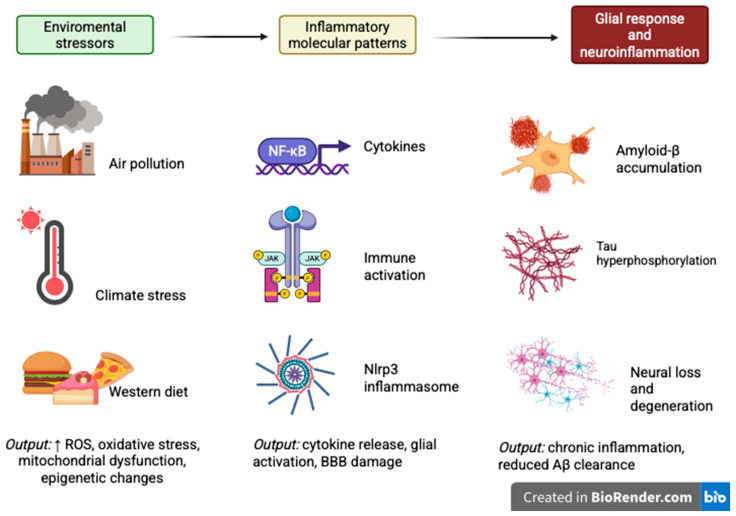
Conceptual model of the “environment–neuroinflammation–Alzheimer’s disease axis.” Environmental stressors, including pollution, extreme temperatures, and a Western diet, lead to oxidative stress and mitochondrial dysfunction, in turn activating NF-κB, JAK/STAT, MAPK, and NLRP3 pathways in glial cells. These cascades perpetuate neuroinflammation, impair amyloid-β clearance, promote tau hyperphosphorylation, and lead to neuronal death and cognitive decline. Genetic and socioeconomic factors may confer added risk. (Created in https://BioRender.com).

**Table 1 cimb-47-00959-t001:** Main signaling pathways involved in neuroinflammation.

Pathway	Key Molecules	Main Functions	Implication in AD	Key References
NF-κB	p65 (RelA), IκB, TNF-α, IL-1β	Regulates pro-inflammatory cytokines, apoptosis, and immune responses	Chronic activation promotes neuroinflammation and neuronal death	[28]
JAK/STAT	JAK1/2, STAT1/3, SOCS	Mediates cytokine signaling and immune cell differentiation	Dysregulation promotes microglial activation and neurotoxicity	[41]
MAPK	ERK, JNK, p38	Controls cellular stress responses, cytokine production, and apoptosis	Involved in tau phosphorylation and synaptic dysfunction	[42,43]
NLRP3 inflammasome	NLRP3, ASC, Caspase-1, IL-1β, IL-18	Triggers pyroptosis and release of pro-inflammatory cytokines	Chronic activation drives neurodegeneration and cognitive decline	[44,49]

**Table 2 cimb-47-00959-t002:** Interplay between amyloid-β (Aβ), tau pathology, and neuroinflammatory responses. The table outlines the principal pathological triggers, the key molecular mediators and signaling pathways involved, and the downstream effects on neuronal function, highlighting the link between protein aggregation, glial activation, and neurodegeneration.

Stimulus/Lesion	Key Mediators/Pathways	Final Effect on CNS	Key References
Aβ aggregation	NF-κB activation in astrocytes; microglial activation; ROS	Neuronal injury, chronic microglial activation, tau hyperphosphorylation	[106,107]
Tau aggregation	Microglial activation; pro-inflammatory cytokines (IL-1β, TNF-α); NLRP3 inflammasome	Synaptic dysfunction, propagation of neurofibrillary tangles, neuronal death	[108,109]
Chronic glial activation	Sustained cytokine release (IL-6, IL-1β, TNF-α); complement activation	Synaptopathy, impaired neurogenesis, cognitive decline	[110]

**Table 3 cimb-47-00959-t003:** Comparative evaluation of evidence linking environmental stressors to neuroinflammation and Alzheimer’s disease. This table summarizes and ranks the evidence (preclinical, clinical, and epidemiological) discussed in this narrative review. This comparative presentation helps to clarify where further work is needed (e.g., micro-/nanoplastics, randomized human trials).

Environmental Stressor	Preclinical Evidence	Clinical Evidence	Epidemiological Evidence	Main References
**PM_2.5_ mano**	**Strong**—multiple in vitro and animal studies showing neuroinflammation and Aβ/tau alterations.	**Limited**	**Strong**—several longitudinal cohorts and meta-analyses showing an increased AD risk.	[152]
**SO_2_, NO_2_ and O_3_**	**Moderate**—mechanisms of neuroinflammation are described in animal models.	**Limited**	**Limited–Moderate**—less consistent evidence, often in studies evaluating also the concomitant PM exposure.	[65,67,70,129]
**Heavy metals**	**Moderate–Strong**—neurotoxicants associated with oxidative stress, inflammation, and neurodegeneration.	**Limited**	**Moderate**—meta-analyses show higher circulating levels in AD vs. controls.	[154,155,156]
**MPs and NPs**	**Emerging**—recent in vitro and animal data show their brain accumulation and neurological damage.	**Essentially absent**	**Weak**—few preliminary human or ecological studies.	[76,77,78]
**Heat waves/extreme temperatures**	**Moderate**—animal studies have detected neuroinflammation in conditions of thermal stress.	**Limited**	**Moderate**—association between heatwaves, hospitalizations and mortality in patients with dementia.	[83,141,143]
**Diet**	**Strong**—multiple preclinical studies show the association between diet, neuroinflammation, and cognitive impairment.	**Moderate**—some limited-sized interventional studies with biomarkers or cognitive outcomes.	**Moderate**—several observational cohorts focusing on MD and WD.	[131,135,137,138]

## Data Availability

No new data were created or analyzed in this study. Data sharing is not applicable to this article.

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
