# Peer review of "Environmental Stressors and Neuroinflammation: Linking Climate Change to Alzheimer’s Disease"

_cimb, 2025, doi:10.3390/cimb47110959_

Round 1
Reviewer 1 Report
Comments and Suggestions for Authors
General Comments:
The authors present a timely and comprehensive narrative review addressing the critical link between environmental stressors, neuroinflammation, and AD. The topic is of significant interest and aligns well with the scope of the journal. However, to enhance the impact and clarity of the review and make it suitable for publication, several major issues need to be addressed. The primary concerns revolve around strengthening the narrative flow, balancing the depth of content, and incorporating a more critical perspective. The connections between the chapters of the article are somewhat abrupt, making it read more like a compilation of independent topics rather than a cohesive narrative. For instance, the transition from the "Neuroinflammation" chapter to the subsequent "Environmental Factors" chapter lacks clear linkage.
Specific Comments by Section:
1. Introduction:
1) There is a missing strong connecting sentence between the discussion of "genetics and environment" and the subsequent detailed explanation of "neuroinflammation." The rationale for delving into neuroinflammation is unclear.
2) The dual role of neuroinflammation (beneficial vs. harmful) is described, but the review's specific focus remains ambiguous. Is the emphasis on the detrimental role, the beneficial role, or both? The central viewpoint should be stated explicitly.
3) The statement of objectives could be more specific and directive.
4) The significance of this review is not sufficiently articulated. It is recommended to explicitly state its contribution to the field.
2. Neuroinflammation Section:
1) The logical flow is unclear. The section introduces signaling pathways, then shifts to microglia and astrocytes, only to return to more signaling pathways. A clearer structure, such as organizing it into: Core Immune Cells (microglia, astrocytes, and their interactions) -> Key Signaling Pathways -> Epigenetic Regulation, would significantly improve clarity.
2) The listed signaling pathways appear somewhat disorganized. Categorizing and integrating them would make the presentation more logical and structured.
3) The statement, "The predominantly activated form, p50-ReIA dimer, is strongly implicated in the detrimental effects of ischemic stroke..." is overly absolute. This dimer has roles in numerous physiological and pathological processes; the phrasing here is too restrictive and inaccurate.
4) The description of microglial M1/M2 phenotypes is outdated. The current scientific consensus views microglial activation as a continuum rather than a simple M1/M2 dichotomy. Using terms like "pro-inflammatory" and "anti-inflammatory" might be more appropriate to reflect the complex microenvironment.
5) The sentence, "These signaling cascades—particularly NF-κB, JAK/STAT, and NLRP3—do not operate in isolation," is an excellent transition. However, it loses impact as a paragraph conclusion. It should be leveraged to introduce the core viewpoint of the subsequent text.
3. Neuroinflammation, Climate Change and Environment Section:
1) The paragraphs jump from one pollutant to another without effective transitions (e.g., from O₃ to SO₂ to NOx). While these are related, the logic feels fractured. It is recommended to establish a clear classification system at the beginning of the chapter to organize the environmental factors.
2) For most studies cited, the presentation is limited to "Exposure A leads to Outcome B," lacking a critical appraisal of the evidence strength. For instance, to what extent can animal model results be extrapolated to humans? Have confounding factors in epidemiological studies been adequately considered?
3) The previous section introduced cells and signals of neuroinflammation, but this connection is weak in this chapter. Many parts merely state that a pollutant "causes" neuroinflammation or cognitive deficits without clearly elucidating how it does so mechanistically.
4) The image quality of Figure 1 is poor and needs to be replaced with a high-resolution version.
4. Neuroinflammation in Alzheimer's Disease Section:
1) The chapter opens by directly describing the neuropathological features of AD. While accurate, this fails to serve as a "bridging" element, leaving the reader unsure of the chapter's specific aim within the broader argument.
2) The introduction of GWAS-identified risk genes lacks depth. The explanation of how these genes mechanistically link neuroinflammation to AD pathology is insufficient. Their specific relationship to AD core processes needs clarification.
3) The transition from discussing microglia and astrocytes to introducing B cells, T cells, and NK cells is very abrupt. There is no explanation for why these peripheral immune cells are relevant in the context of a central nervous system disease.
5. Alzheimer's Disease and Environment Section:
1) It is recommended to introduce a clear classification and narrative framework at the beginning of the chapter to outline its organizational logic for the reader.
6. Conclusions and Future Perspectives Section:
1) The current conclusion loosely restates the findings of each chapter. It should first distill a unified, powerful core message and directly articulate the central argument of the review.
2) The outlook section should present novel ideas or propose deeper mechanistic investigations, moving beyond a generic "to-do" list.
Author Response
Roma, November 13rd 2025
Dear Editor of Current Issue In Molecular Biology
First, my coauthors and I would like to thank you sincerely for this opportunity to cooperate. We profoundly thank the reviewers for the comments and useful suggestions to improve the paper.
This is a point-by-point list of changes made in the paper:
REVIEWER 1
General Comments:
The authors present a timely and comprehensive narrative review addressing the critical link between environmental stressors, neuroinflammation, and AD. The topic is of significant interest and aligns well with the scope of the journal. However, to enhance the impact and clarity of the review and make it suitable for publication, several major issues need to be addressed. The primary concerns revolve around strengthening the narrative flow, balancing the depth of content, and incorporating a more critical perspective. The connections between the chapters of the article are somewhat abrupt, making it read more like a compilation of independent topics rather than a cohesive narrative. For instance, the transition from the "Neuroinflammation" chapter to the subsequent "Environmental Factors" chapter lacks clear linkage.
Specific Comments by Section:
- Introduction:
1) There is a missing strong connecting sentence between the discussion of "genetics and environment" and the subsequent detailed explanation of "neuroinflammation." The rationale for delving into neuroinflammation is unclear.
We have added a connecting phrase as requested.
2) The dual role of neuroinflammation (beneficial vs. harmful) is described, but the review's specific focus remains ambiguous. Is the emphasis on the detrimental role, the beneficial role, or both? The central viewpoint should be stated explicitly.
We have clarified the focus of the review by explicitly stating that our discussion primarily centers on the detrimental and chronic aspects of neuroinflammation, which link environmental stressors to neurodegeneration and Alzheimer’s disease. Nevertheless, we also acknowledge the physiological and transient protective functions of neuroinflammatory responses, as they provide important context for understanding how these mechanisms become maladaptive over time.
3) The statement of objectives could be more specific and directive.
We have revised the Introduction to provide a clearer and more explicit statement of the review’s objectives.
4) The significance of this review is not sufficiently articulated. It is recommended to explicitly state its contribution to the field.
We have explained the meaning and importance of the revision in a clear and detailed manner.
- Neuroinflammation Section:
1) The logical flow is unclear. The section introduces signaling pathways, then shifts to microglia and astrocytes, only to return to more signaling pathways. A clearer structure, such as organizing it into: Core Immune Cells (microglia, astrocytes, and their interactions) -> Key Signaling Pathways -> Epigenetic Regulation, would significantly improve clarity.
We have reorganized the section as requested.
2) The listed signaling pathways appear somewhat disorganized. Categorizing and integrating them would make the presentation more logical and structured.
We have modified the text.
3) The statement, "The predominantly activated form, p50-ReIA dimer, is strongly implicated in the detrimental effects of ischemic stroke..." is overly absolute. This dimer has roles in numerous physiological and pathological processes; the phrasing here is too restrictive and inaccurate.
We have modified the sentence as requested.
4) The description of microglial M1/M2 phenotypes is outdated. The current scientific consensus views microglial activation as a continuum rather than a simple M1/M2 dichotomy. Using terms like "pro-inflammatory" and "anti-inflammatory" might be more appropriate to reflect the complex microenvironment.
We have modified the text as requested.
5) The sentence, "These signaling cascades—particularly NF-κB, JAK/STAT, and NLRP3—do not operate in isolation," is an excellent transition. However, it loses impact as a paragraph conclusion. It should be leveraged to introduce the core viewpoint of the subsequent text.
We have modified the text.
- Neuroinflammation, Climate Change and Environment Section:
1) The paragraphs jump from one pollutant to another without effective transitions (e.g., from O₃ to SO₂ to NOx). While these are related, the logic feels fractured. It is recommended to establish a clear classification system at the beginning of the chapter to organize the environmental factors.
We have introduced categorized environmental factors into three main groups: (1) air pollutant variables, (2) climate-related physical stressors, and (3) dietary and lifestyle factors. Furthermore, paragraph flow was amended to aid in logical reasoning through an iteration of subsections. These proceeds improved readability and logically smoothed the conceptual transition.
2) For most studies cited, the presentation is limited to "Exposure A leads to Outcome B," lacking a critical appraisal of the evidence strength. For instance, to what extent can animal model results be extrapolated to humans? Have confounding factors in epidemiological studies been adequately considered?
We have inserted explicit remarks throughout the section highlighting the translational limitations of animal models, potential confounders in epidemiological evidence, and the need for standardized biomarkers and longitudinal data to establish causality.
3) The previous section introduced cells and signals of neuroinflammation, but this connection is weak in this chapter. Many parts merely state that a pollutant "causes" neuroinflammation or cognitive deficits without clearly elucidating how it does so mechanistically.
We have now revised the section to explicitly connect each environmental factor with key neuroinflammatory pathways (e.g., NF-κB, JAK/STAT, MAPK, NLRP3) and glial cell responses (microglial activation, astrocyte–microglia crosstalk, oxidative stress). These revisions clarify how pollutants and climate-related stressors mechanistically trigger or amplify neuroinflammation and neurodegeneration.
4) The image quality of Figure 1 is poor and needs to be replaced with a high-resolution version.
We have completely redesigned the figure in accordance with the other revisions.
- Neuroinflammation in Alzheimer's Disease Section:
1) The chapter opens by directly describing the neuropathological features of AD. While accurate, this fails to serve as a "bridging" element, leaving the reader unsure of the chapter's specific aim within the broader argument.
We've added a short introductory paragraph to the beginning of this section, explicitly linking general neuroinflammatory mechanisms (Section 2) and environmental triggers (Section 3) to their relevance in Alzheimer's disease. This addition clarifies the purpose of the section: to illustrate how the previously described molecular and environmental pathways converge in the specific context of AD pathophysiology.
2) The introduction of GWAS-identified risk genes lacks depth. The explanation of how these genes mechanistically link neuroinflammation to AD pathology is insufficient. Their specific relationship to AD core processes needs clarification.
We have expanded this section to clarify how key AD susceptibility genes, such as TREM2, CD33, CR1, CLU, ABCA7, and APOE, functionally link neuroinflammation to Alzheimer's pathology.
3) The transition from discussing microglia and astrocytes to introducing B cells, T cells, and NK cells is very abrupt. There is no explanation for why these peripheral immune cells are relevant in the context of a central nervous system disease.
We have added a bridging paragraph to clarify that systemic immune responses can influence the CNS through blood–brain barrier alterations, peripheral cytokine signaling, and immune cell infiltration, particularly under neurodegenerative and aging-related conditions.
- Alzheimer's Disease and Environment Section:
1) It is recommended to introduce a clear classification and narrative framework at the beginning of the chapter to outline its organizational logic for the reader.
We have added a short introductory paragraph at the beginning of this section, outlining three major categories of environmental risk factors—(1) air pollutants and toxins, (2) lifestyle and dietary factors, and (3) climate-related physical stressors—and explaining how each category interacts with the neuroinflammatory mechanisms previously described.
- Conclusions and Future Perspectives Section:
1) The current conclusion loosely restates the findings of each chapter. It should first distill a unified, powerful core message and directly articulate the central argument of the review.
We have rewritten the opening of the Conclusions and Future Perspectives section to clearly articulate the unifying message of the manuscript.
2) The outlook section should present novel ideas or propose deeper mechanistic investigations, moving beyond a generic "to-do" list.
We agree that the Outlook section should not merely summarize research needs but rather outline new conceptual and mechanistic directions. We have therefore revised the final part of the section to propose emerging lines of investigation, including the integration of multi-omics and single-cell approaches to map inflammatory cell states, systems-level modeling of neuroimmune interactions under environmental stress, and cross-disciplinary frameworks combining neuroscience, immunology, and environmental health.
We thank You for your constructive critique and we hope the review process has led to an improved manuscript.
If additional changes are warranted, we will make them.
We hope that this revised version of our manuscript may now be found suitable for publication.
Sincerely,
Rossella Cianci
Reviewer 2 Report
Comments and Suggestions for Authors
The manuscript "Environmental Stressors and Neuroinflammation: Linking Climate Change to Alzheimer’s Disease” discusses an emerging topic with significant scientific and global health relevance: the connection between environmental factors, neurodegeneration, and climate change. The introduction, molecular mechanisms, environmental influences, implications in Alzheimer's disease, and conclusions are all logically organized in the manuscript to make comprehension simpler. Additionally, the authors link environmental exposures and food to inflammatory pathways (NF-κB, JAK/STAT, and NLRP3) by integrating epidemiological, molecular, and genetic evidence. The authors indicate an attempt to keep scientific currency by citing recent findings from 2023 to 2025, such as GWAS and Lancet Countdown. Additionally, a topic that did not receive adequate coverage in previous reviews, the manuscript offers an in-depth description of the mechanisms of neuroinflammation and the way they correlate to environmental stress.
However, the current manuscript should be improved, following the suggestions below:
- The following articles should be reviewed and cited by authors:
- Casella, C., Cornelli, U., Ballaz, S., Zanoni, G., Merlo, G., & Ramos-Guerrero, L. Plastic Smell: A Review of the Hidden Threat of Airborne Micro and Nanoplastics to Human Health and the Environment.Toxics, 2025, 13(5), 387.https://doi.org/10.3390/toxics13050387
Airborne microplastics (MPs), a new pollutant identified by the WHO and UNEP, are not covered in the present review, which focuses on traditional pollutants including PM₂.₅, NO₂, O₃, and SO₂. This reference expands the range of environmental stressors and brings the review into line with the most recent global environmental health priorities. MPs/NPs can be breathed, penetrate the blood-brain barrier, and cause neuroinflammation, mitochondrial dysfunction, and oxidative stress—the same pathogenic processes that are covered in the article (NF-κB, NLRP3, ROS), according to recent studies. Consequently, the inclusion of this citation supports the main contention that diverse environmental influences converge on shared inflammatory pathways. This directly affects public health and is consistent with the manuscript's suggestions for lowering environmental exposure and lowering the risk of neurodegenerative diseases.
- The review is mostly descriptive; it does not critically evaluate or compare research that are in disagreement. The results of the cited studies are provided without any indication of their design, limitations, or level of evidence. A comparative table or section ranking the evidence (preclinical, clinical, and epidemiological) would be helpful.
- There are numerous minor syntax and grammar errors (inconsistent use of articles, prepositions, and punctuation):
- “both genetical and environmental risk factors” → both genetic and environmental risk factors
- “activation produces two different mechanisms, which end in the involvement of an unparalleled entity into the cuing mechanism” → unclear phrasing; requires rewording for precision.
- Despite collecting from an abundance of studies, the authors do not present a novel conceptual hypothesis or unique visual model that combines the pathways (such as the "environmental stressor–neuroinflammation–AD axis"). A conceptual framework or integrative figure summarizing the relationships between inflammatory cascades, pollution, diet, and climate would be helpful for the authors.
- Some acronyms, like AD and NF-κB, are introduced before being defined.
- Some references are included without providing an explanation (e.g., [17] or [37–39]).
- There are no obvious cross-references in the text of the figures and tables ("Figure 1" could be more visually explanatory).
- It is unclear if the review was registered or if the studies chosen were biased.
- The final section mentions preventive and therapeutic strategies, but only superficially and without feasibility analysis, limitations, or relevant clinical trials. I advise the authors to delve deeper into the viability of these strategies (e.g., NLRP3 inhibitors, anti-inflammatory diets, public policies).
- I suggest that the authors answer the following questions, which are useful for deepening or expanding the manuscript:
- Could you clarify on the investigated paper's inclusion and exclusion criteria?
- Does the extent of the risk connected to environmental exposure and neurodegeneration have quantitative data (meta-analysis) to support it?
- Which biomarkers do the researchers think have the best chance of detecting environmental damage in Alzheimer's disease?
- Based on available data, what is the relationship between environmental influences and genetic determinants (such APOE4)?
- Would you be able to provide a conceptual model that summarizes the mechanistic relationships, as described?
The manuscript needs significant editing in terms of style, methodology, and critical analysis, yet it is thematically strong and extremely relevant. The present review would have more impact if the methodology were made clearer, the evidence given priority, and the language improved, even though the figures and tables are helpful. After major revision, it might make a significant addition to Current Issues in Molecular Biology.
Author Response
Rome, November 13rd 2025
Dear Editor of Current Issue In Molecular Biology
First, my coauthors and I would like to thank you sincerely for this opportunity to cooperate. We profoundly thank the reviewers for the comments and useful suggestions to improve the paper.
This is a point-by-point list of changes made in the paper:
REWIEVER 2
The manuscript "Environmental Stressors and Neuroinflammation: Linking Climate Change to Alzheimer’s Disease” discusses an emerging topic with significant scientific and global health relevance: the connection between environmental factors, neurodegeneration, and climate change. The introduction, molecular mechanisms, environmental influences, implications in Alzheimer's disease, and conclusions are all logically organized in the manuscript to make comprehension simpler. Additionally, the authors link environmental exposures and food to inflammatory pathways (NF-κB, JAK/STAT, and NLRP3) by integrating epidemiological, molecular, and genetic evidence. The authors indicate an attempt to keep scientific currency by citing recent findings from 2023 to 2025, such as GWAS and Lancet Countdown. Additionally, a topic that did not receive adequate coverage in previous reviews, the manuscript offers an in-depth description of the mechanisms of neuroinflammation and the way they correlate to environmental stress.
However, the current manuscript should be improved, following the suggestions below:
The following articles should be reviewed and cited by authors:
Casella, C., Cornelli, U., Ballaz, S., Zanoni, G., Merlo, G., & Ramos-Guerrero, L. Plastic Smell: A Review of the Hidden Threat of Airborne Micro and Nanoplastics to Human Health and the Environment.Toxics, 2025, 13(5), 387.https://doi.org/10.3390/toxics13050387
Airborne microplastics (MPs), a new pollutant identified by the WHO and UNEP, are not covered in the present review, which focuses on traditional pollutants including PM₂.₅, NO₂, O₃, and SO₂. This reference expands the range of environmental stressors and brings the review into line with the most recent global environmental health priorities. MPs/NPs can be breathed, penetrate the blood-brain barrier, and cause neuroinflammation, mitochondrial dysfunction, and oxidative stress—the same pathogenic processes that are covered in the article (NF-κB, NLRP3, ROS), according to recent studies. Consequently, the inclusion of this citation supports the main contention that diverse environmental influences converge on shared inflammatory pathways. This directly affects public health and is consistent with the manuscript's suggestions for lowering environmental exposure and lowering the risk of neurodegenerative diseases.
We have incorporated and discussed the article by Casella et al. (2025) as requested in the section “Neuroinflammation, climate change and environment” where MPs and NPs are described.
The review is mostly descriptive; it does not critically evaluate or compare research that are in disagreement. The results of the cited studies are provided without any indication of their design, limitations, or level of evidence. A comparative table or section ranking the evidence (preclinical, clinical, and epidemiological) would be helpful.
We added a comparative evidence table (Table 3) that ranks evidence (preclinical, clinical, epidemiological). This table, together with other critical observations added throughout the text, addresses the request for a more analytical rather than descriptive approach.
There are numerous minor syntax and grammar errors (inconsistent use of articles, prepositions, and punctuation):
“both genetical and environmental risk factors” → both genetic and environmental risk factors
“activation produces two different mechanisms, which end in the involvement of an unparalleled entity into the cuing mechanism” → unclear phrasing; requires rewording for precision.
An expert author revised the manuscript to correct syntax, grammar, and phrasing inconsistencies. Ambiguous expressions have been rewritten.
Despite collecting from an abundance of studies, the authors do not present a novel conceptual hypothesis or unique visual model that combines the pathways (such as the "environmental stressor–neuroinflammation–AD axis"). A conceptual framework or integrative figure summarizing the relationships between inflammatory cascades, pollution, diet, and climate would be helpful for the authors.
In the revised version, we have described a conceptual model (new Figure 1) that integrates the main inflammatory cascades with major environmental stressors (air pollution, heat exposure, diet, and emerging pollutants).
Some acronyms, like AD and NF-κB, are introduced before being defined.
We have explained these acronyms as requested.
Some references are included without providing an explanation (e.g., [17] or [37–39]).
We have provided an explanation to the indicated references.
There are no obvious cross-references in the text of the figures and tables ("Figure 1" could be more visually explanatory).
We better connected the figure and tables with the text and we have redesigned Figure 1 as you requested in your previous comment
It is unclear if the review was registered or if the studies chosen were biased.
As requested, we better explained the type of review, which is not registered, in the introduction.
The final section mentions preventive and therapeutic strategies, but only superficially and without feasibility analysis, limitations, or relevant clinical trials. I advise the authors to delve deeper into the viability of these strategies (e.g., NLRP3 inhibitors, anti-inflammatory diets, public policies).
We expanded the final section as requested.
I suggest that the authors answer the following questions, which are useful for deepening or expanding the manuscript:
Could you clarify on the investigated paper's inclusion and exclusion criteria?
We clarify the methodology of study selection in the introduction.
Does the extent of the risk connected to environmental exposure and neurodegeneration have quantitative data (meta-analysis) to support it?
We have added quantitative data and meta-analysis throughout the text, where available, for example in the section “Alzheimer’s disease and environment”.
Which biomarkers do the researchers think have the best chance of detecting environmental damage in Alzheimer's disease?
We have added a paragraph on emerging biomarkers in the literature to the conclusions and future prospects section.
Based on available data, what is the relationship between environmental influences and genetic determinants (such APOE4)?
Would you be able to provide a conceptual model that summarizes the mechanistic relationships, as described?
We provided a conceptual model to summarizes the described pathways, which is the “environment-neuroinflammation-AD axis”, as described in the text and clarified by the new figure.
We thank You for your constructive critique and we hope the review process has led to an improved manuscript.
If additional changes are warranted, we will make them.
We hope that this revised version of our manuscript may now be found suitable for publication.
Sincerely,
Rossella Cianci
Round 2
Reviewer 1 Report
Comments and Suggestions for Authors
agree to publish
Reviewer 2 Report
Comments and Suggestions for Authors
Dear Editor,
Dear authors,
After making the necessary changes based on my suggestions, I believe the paper ”Environmental Stressors and Neuroinflammation: Linking Climate Change to Alzheimer’s Disease" is ready for publication in your prestigious journal.